# Differential Ground-Based Radar Interferometry for Slope and Civil Structures Monitoring: Two Case Studies of Landslide and Bridge

**Jiyuan Hu [1,2,3], Jiming Guo [1,*] , Yi Xu [1], Lv Zhou [1], Shuai Zhang [1] and Kunfei Fan [4]**

[1] School of Geodesy and Geomatics, Wuhan University, Wuhan 430079, China; plgk@whu.edu.cn (J.H.); dhxzyz@whu.edu.cn (Y.X.); zhoulv@whu.edu.cn (L.Z.); 2016102140015@whu.edu.cn (S.Z.)

[2] Helmholtz Centre Potsdam, GFZ German Research Centre for Geosciences, 14473 Potsdam, Germany

[3] Guangxi Key Laboratory of Spatial Information and Geomatics, Guilin University of Technology, Guilin 541004, China

[4] Nanning Natural Resources Information Center, Nanning 530022, China; kunfei_f@sina.com

* Correspondence: jmguo@sgg.whu.edu.cn; Tel.: +86-133-0862-0798

**Abstract:** Ground-based radar interferometry, which can be specifically classified as ground-based synthetic aperture radar (GB-SAR) and ground-based real aperture radar (GB-RAR), was applied to monitor the Liusha Peninsula landslide and Baishazhou Yangtze River Bridge. The GB-SAR technique enabled us to obtain the daily displacement evolution of the landslide, with a maximum cumulative displacement of 20 mm in the 13-day observation period. The virtual reality-based panoramic technology (VRP) was introduced to illustrate the displacement evolutions intuitively and facilitate the following web-based panoramic image browsing. We applied GB-RAR to extract the operational modes of the large bridge and compared them with the global positioning system (GPS) measurement. Through full-scale test and time-frequency result analysis from two totally different monitoring methods, this paper emphasized the 3-D display potentiality by combining the GB-SAR results with VRP, and focused on the detection of multi-order resonance frequencies, as well as the configure improvement of ground-based radars in bridge health monitoring.

**Keywords:** GB-SAR; GB-RAR; VRP; displacement evolution; operational modes

## 1. Introduction

Spaceborne synthetic aperture radar (SAR) systems, from the pioneer SEASAT to the present Sentinel-1 satellite, provide a powerful tool for deformation measurement with millimeter accuracy by adopting differential interferometry or multi-temporal interferometry [1,2]. Thus, SAR systems have been used in many application fields, e.g., surface subsidence [3–5], earthquakes [6–8], and landslides [9,10]. Where the long-term and large-scale with slow deformation areas are characterized, the use of spaceborne sensors has been shown to be successful and effective. Meanwhile, in some complimentary, cases, e.g., individual structures, local landslides, steep slopes, and scenes with fast deformation rates involving temporal decorrelation, an alternative strategy arises from the use of ground-based radar interferometry [11].

Ground-based radar interferometry system (GBRI) is the realization and effective complementarity of spaceborne radar system on the ground. It emits and receives a burst of microwaves to remotely detect small displacements of targets through phase differentials [4,12]. Based on the image resolution/antenna physical dimension, a specific class of ground-based radar interferometry systems can be classified into: (1) Real aperture radars (SARs), such as GAMMA's Portable Radar Interferometer (GPRI) and Image By Interferometric Survey-Structure (IBIS-S); and (2) Synthetic aperture radar (SARs), such as the

linear synthetic aperture radar (LISA) and Image By Interferometric Survey-Landslide (IBIS-L). For the ground-based synthetic aperture radar (GB-SAR), its antenna is synthetically elongated by moving the radar along a rail track perpendicular to the look direction. In the range direction, it uses the stepped frequency continuous wave (SFCW) or frequency modulated continuous wave (FMCW) technique to implement sampling in frequency domain, which can realize the two-dimensional imaging capability with range and cross-range resolution of a large scene [12–14]. Thus, GB-SAR, as a supplementary of spaceborne satellite, is widely used in landslides [15,16], mines [17], sinkhole subsidence [18], glaciers [19], and volcanoes [20]. Unlike GB-SAR, the ground-based real aperture radar (GB-RAR), with a radar sensor mounted on a tripod transmitting and receiving pulses of energy, can only obtain the range resolution but with a higher sampling rate [13,21]. Thus, the applications of GB-RAR are mainly in manmade structures, such as bridges [22,23], buildings [24,25], dams [26], towers [27], and chimneys [20].

Although existing literatures have shown GB-SARs and GB-RARs as powerful tools for extracting the two-dimensional displacement evolution of a large scene or dynamic vibration parameters of a manmade structure, there is still room for improvement in the interpretation and exhibition of these GBRI-derived results, on-surveying, and mapping professionals. For those modal shape parameters extracted from frequency or time domain methods, most studies have emphasized the methods of modal-parameter identification and validation using different sensors dataset, but have neglected further discussion of the modal assurance criterion diagrams (MAC) between multiple mode shapes that is sensitive to damage scenarios examination, e.g., Gentile et al. [21], Stabile et al. [22], Hu et al. [24], Luzi et al. [25], and Xi et al. [28]. In view of the first issue, the virtual reality-based panoramic technology (VRP) is implemented to the panorama images generation and spherical projection of the study area images collected by an unmanned aerial vehicle (UAV). Subsequently, the spherical panoramic images are embedded into the web to realize the panoramic roaming and map navigation. Most of the scene deformation results were shown in a displacement color bar-adhered two-dimensional bitmap with or without scene geomorphology superposition, e.g., Tarchi et al. [15], and Luzi et al. [29], Leva et al. [30]. Thus, it is difficult to match them with the specific scene location. The in-situ measurement of high-frequency GPS (10 Hz) and the no-contact remote sensing method of GB-RAR (17 Hz) were conducted on the Wuhan Baishazhou Yangtze River Bridge, and the dynamic vibration parameters were extracted using the enhanced frequency domain decomposition technique (EFFD) and sequential quadratic programming-based venetic algorithm (SQP-GA). Through these damage-sensitive features, ambient vibration testing (AVT)-based damage examination features of the bridge were presented.

The paper is organized as follows: Section 2 briefly recalls the working principle of GB-SAR, GB-RAR, VRP, and the adopted method for modal parameters identification. Section 3 provides a description of the two experimental campaigns, underlining the data acquisition strategy and preliminary results. Section 4 presents the results from VRP and Web-based panoramic image browsing and identifies modal parameters and MAC results, followed by several concluding remarks related to the integration of multi-sensor and interdisciplinary disciplines on large-scene and individual structures.

## 2. Working Principle of GBRI, VRP, and Method for Modal Parameters Identification and Discrimination

Although there are many kinds of ground-based radar interferometry systems on the market, e.g., linear SAR [23], rotary SAR [31], ArcSAR [32], multiple input multiple output SAR [33], and moving slot [34], they are different in instrument volumes and measurement ways. The principle of acquiring target displacement is essentially realized by the stepped frequency(SF)/frequency modulation (FM) continuous wave (CW) and interferometry technique. In this paper, we only focused on the linear SAR system and used the IBIS-L and IBIS-S developed by Ingegneria dei Sistemi S.p.A. (IDS) in collaboration with the Department of Electronics and Telecommunication of the Florence University as an example for illustrations.

## 2.1. Principles of GBRI

The technique in range direction integrates a Stepped Frequency Continuous Wave (SF-CW) synthesizer to continuously emit a set of sweeps containing a SF signal with bandwidth $B$, a frequency step size $\Delta f$ (Figure 1), and a receiver to collect the energy reflected by the illuminated objects [35]. Thereby, it can achieve ultra-bandwidth $B$ ($B = (N-1)\Delta f$), thus leading to a higher range resolution $\delta_r$ and reducing the instantaneous bandwidth requirement of digital signal processor [36].

$$\delta_r = \frac{c\tau}{2} = \frac{c}{2B} \tag{1}$$

where $\tau$ is the pulse width, allowing an indicator of two points illuminated in range by the instrument to be distinguished, and $c$ is the speed of light.

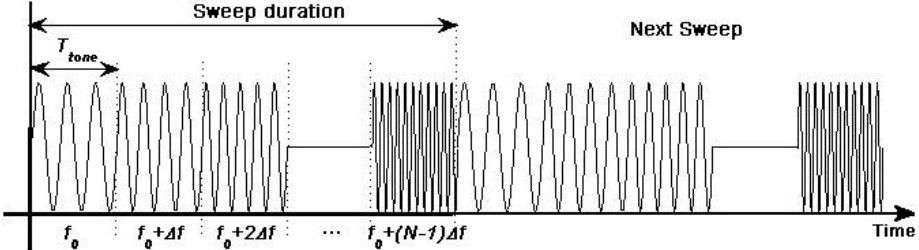

**Figure 1.** Representation of the SFCW waveform in time and the frequency domain.

The cross-range resolution is obtained without using long antennas by the SAR technique, i.e., the sensor moves in a "stop and go" mode, scans scene at each step, and thus acquires mono-dimensional images which are subsequently compressed (image focusing) in a single two-dimension complex images using the inverse discrete Fourier transform (IDFT) [36,37]. Thereby, the achievable cross-range angular step $\delta_\theta$ and its corresponding cross-range resolution $\delta_{rc}$ are given:

$$\delta_\theta = \frac{\lambda}{2L} \ , \ \delta_{rc} = \frac{\lambda}{2L}r \tag{2}$$

The final complex images (comprising amplitude $A$ and phase $\varphi$), where each pixel contains the in-phase ($I$) and the quadrature ($Q$) components, and the polar image geometry (Figure 2) with a constant range sampling step of $\delta_r$ and a constant azimuth angular step of $\delta_\theta$, are given by Equations (3) and (4), respectively.

$$\varphi = \tan^{-1}\left(\frac{Q}{I}\right) , \ A = \sqrt{I^2 + Q^2} \tag{3}$$

$$r = y * \delta_r \ , \ \beta = (x - x_c) * \delta_\theta \tag{4}$$

where $\lambda$ is the wavelength, $L$ is the length of the rail, and $r$ and $\beta$ are the radar to pixel distance the off-central column angle, which can be calculated from a given pixel ($x,y$).

Once the time domain responses (amplitude and phase) are determined, the phase discrepancy during a time interval of the same range bin $\Delta\varphi$ is obtained by complex conjugate multiplication of the two echoes, expressed in Equation (5), which directly reflect the changes in distance between this pixels and the radar, i.e., radial displacement $\Delta d$ in line of sight (LOS), expressed in Equation (6).

$$\Delta\varphi = \varphi_{t1}(x,y)\varphi_{t1}(x,y)^* = \Delta\varphi_{disp} + \Delta\varphi_{atm} + \Delta\varphi_{noise} - 2\pi n \tag{5}$$

$$\Delta d = \frac{\lambda}{4\pi}\Delta\varphi_{disp} = \frac{\lambda}{4\pi}(\Delta\varphi - \Delta\varphi_{atm} - \Delta\varphi_{noise} + 2\pi n) \tag{6}$$

where $\Delta\varphi$ is the sum phase of three parts: Displacement, with which are are concerned; atmosphere, which, for a small area, can be eliminated from surrounding stable points; and the

thermal/environmental noise practically, which is removed by selected points with high signal to noise ratio (SNR) [11,15,35].

Except for the lack of SAR capability, GB-RAR, like IBIS-S, shares the same range imaging and interferometry techniques with the above-mentioned GB-SAR. The tripod replaces the rail, while the range sampling rates are up to 200 Hz with a displacement detection accuracy of 0.01 mm [24], which is especially suitable for monitoring the dynamic vibration of structures.

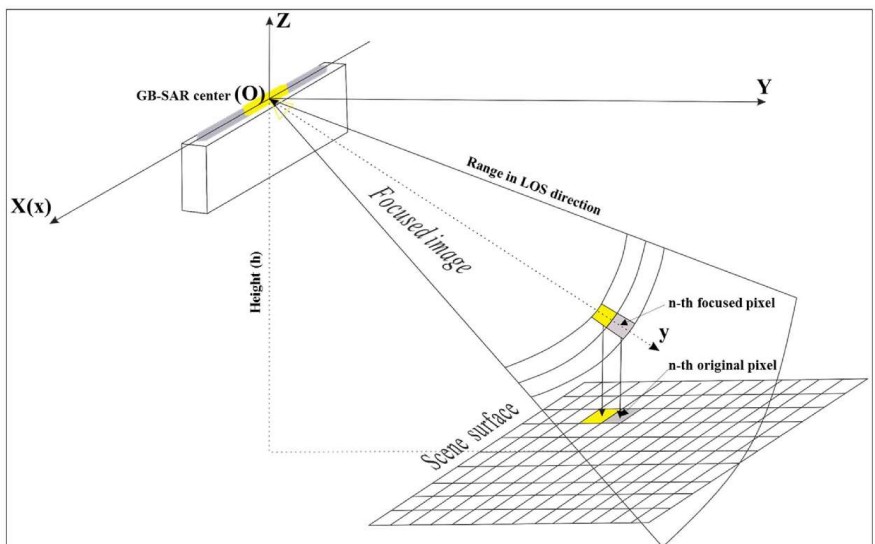

**Figure 2.** Graphical sketch of image geometry and focusing algorithm. The O-xy is the 2-D radar coordinate system, which can be transformed into 3-D Cartesian coordinate O-XYZ according to the image geometry.

### 2.2. VR-based Panoramic Technology of Web

Virtual reality based the Panorama of Web technique (VRPW), also known as three-dimensional panoramic virtual reality, is a real scene virtual reality technology based on panoramic images [38,39]. This technique mainly focuses on the accuracy of virtual environment representation, the authenticity of virtual environment perception information synthesis, the naturalness of human-virtual environment interaction, real-time display, graphics generation, intelligent technology, and other issues, allowing users to perceive the virtual environment personally, so as to achieve the goal of exploring and understanding objective things.

The steps of creating the Liusha Peninsula landslide (the study area described in Section 3.1) virtual scene with the technology of the VRP were as follows: (1) The Rotor UAV hovered at fixed point in the air with a camera and took common-view projection photography of the landslide area; (2) Panorama image generation, which mainly included images of spherical projection, as well as mosaic and sky-filling decoration. The planar graph, with a ratio of length to height of 2:1, was obtained after using the latitude–longitude mapping method; (3) The 360-degree panoramic images, re-spherically projected from the two-dimensional picture, were embed into the webpage for users to sweep, revolve, switch, zoom out, and zoom in among different scenes using only the mouse, allowing users to feel like they were in the virtual scene. On this basis, based on the 3-D development engine Unity®, the virtual reality scene was built by means of objects and components. Finally, the deformation monitoring results were imported and published as the objects in Unity scene to realize the establishment of high visual deformation model.

### 2.3. Modal Parameters Identification

Enhanced frequency domain decomposition (EFFD) is an improved method proposed by Brincker et al. [40] on the basis of frequency domain decomposition, which is widely applied to modal identification of large structures under environmental excitation. Unlike the frequency domain

decomposition method, in this technique, the decomposed single-degree-of-freedom power spectral density (PSD) function is inversely Fourier transformed to obtain the time-domain correlation function. Then, the logarithmic attenuation method is used to calculate the frequency and damping ratio [41,42]. The principle in the EFFD technique is easily understood by recalling that any response can be expressed with unknown input signal and output signal in modal coordinates:

$$G_{yy}(jw) = H^*(jw)G_{xx}(jw)H^T(jw) \tag{7}$$

$G_{yy}(jw) \in \mathbb{R}^{m \times m}$ PSD matrix of responses, where $m$ is the number of responses; $G_{xx}(jw) \in \mathbb{R}^{l \times l}$ PSD matrix of the input, where $l$ is the number of inputs; $H(jw) \in \mathbb{R}^{m \times l}$ matrix of frequency response function (FRF); $*$ and superscript $T$ denote complex conjugate and transpose.

In a multi-degree-of-freedom system, the frequency response function can be written in the following partial fractions and forms:

$$H(jw) = \sum_{i=1}^{n} \left[ \frac{R_i}{(jw - \lambda_i)} + \frac{R_i^*}{(jw - \lambda_i^*)} \right] \tag{8}$$

where $i$ is the order of mode shape, $R_i$ is the $i$−th matrix of FRF, $\lambda_i$ is the eigenvalues of discrete systems, and $w$ is the natural frequency. Their relation with $\lambda_i$ can be expressed as:

$$\lambda_i = -\xi w_i + j \sqrt{1 - \xi^2} w_i \tag{9}$$

Thus, the estimation of the output $\hat{G}_{yy}(jw)$, known at discrete frequencies $w_i$, is then decomposed by taking the singular value decomposition (SVD) of the PSD matrix,

$$\hat{G}_{yy}(jw) = U_i S_i U_i^H \tag{10}$$

where $S_i$ denotes the diagonal matrix holding the scalar singular values, and $U_i$ is the unitary matrix holding the singular vectors, which indicates the mode shape of structure.

Furthermore, a hybrid optimization method (SQP-GA) that combines the genetic algorithm with sequential quadratic programming was also adopted in this study to make a cross-comparison with the EFFD-derived results. Detail explication can be found in the study by Hu J.Y et al [24], in which the SQP-GA method was successfully used to extract the modal parameters of a high-rise building.

## 3. The Experimental Campaigns

Two case studies, where the IBIS systems (IBIS-L and IBIS-S) were characterized by the GB-SAR and GB-RAR technique mentioned above, were used to study the deformation evolution of the Liusha Peninsula landslide and the dynamical behavior of the Baishazhou Yangtze River Bridge. The two case studies are reported in this section. In this section, we focused on the outstanding results derived from IBIS for each case. The discussion involved validation, comparison with multi-sensors, and MAC calculation. Detailed operational parameters can be found in the studies by Rödelsperger et al. [13] and Monserrat Hernández [37].

### 3.1. The Liusha Peninsula Landslide

#### 3.1.1. IBIS-L Configuration

The Liusha Peninsula landslide occurred on the Liusha peninsula on the north bank of the Yong River in Nanning, China, with Yinghua Road on the south and Peninsula Community on the east. It first happened on 12 July 2014, due to building loads and heavy rainfall, causing the soil mass migration of an area with width of 100 m, thickness of 15 m, and accumulated soil body of 40,000 cubic meters. Such a large amount of soil mass migration directly led to cracks and tilt in some pillars, beams, and walls of the A6 residential building and its podium near the Yong River (Figure 3).

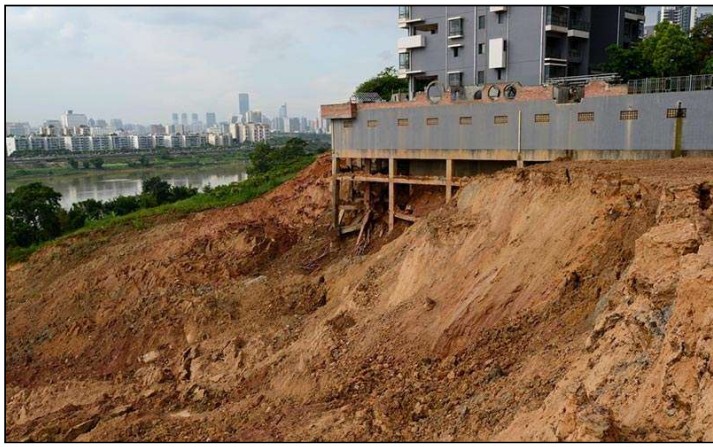

**Figure 3.** Example of the heavy damages on building caused by the Liusha Peninsula landslide on 12 July 2014.

To assess the possibility of further landslides and ensure the safety of the Peninsula Community, a measurement campaign was carried out from the 24 September to 6 October 2017. The IBIS-L instrument was installed at stable a datum point of approximately 420 m on the opposite levee in front of the Liusha Peninsula landslide (Figure 4b). The arrangement of the instrumentation in the field is shown in Figure 4a. The 2-m rail was precisely mounted on a custom steel shelf with five threaded studs positioning holes of 16 mm in diameter on its upper surface. During the whole observation period, the steel frame was fixed on a stable ground, which ensured a millimetric repositioning of the rail and thus avoided a later co-registration of the datasets acquired in different field campaigns [11]. According to the characteristics of short sampling period of GB-SAR and small monitoring scene in this study, four stable and high SNR points located in the northeast and southwest of the central landslide area, respectively, were selected under the assumption of atmospheric homogeneity in the observation scene to remove the atmospheric impact [43,44].

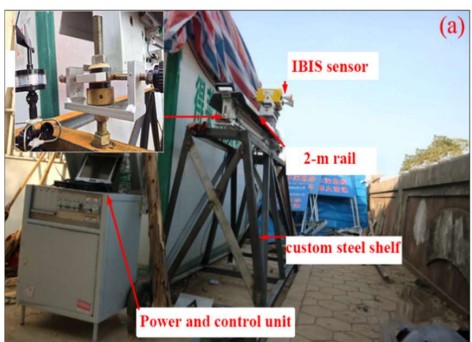
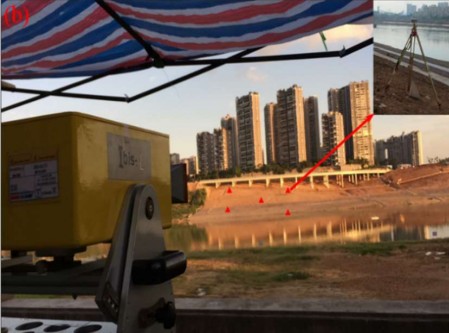

**Figure 4.** (**a**) Field settings and component connections of IBIS-L. The insert in upper left corner is an enlarged view of precise rail positioning. (**b**) Overall view of data acquisition. The red triangles represent the global positioning system (GPS) receivers and the corner reflectors are shown in the upper right corner.

Moreover, we also implemented GPS measurements on 29 September 2017 with five receivers (T1–T5) installed at feature points on the landslide. At each position of GPS receiver, a corner reflector was fixed to facilitate post-analysis (Figure 4b).

3.1.2. Displacement Results

The area for analysis, illuminated by the IBIS-L sensor, extended to approximately 250 m in range and 600 m in width, primarily comprising the landslide, bridge, and some buildings (Figure 4b). Each image scan took roughly six minutes to perform a full SAR measure. The thermal signal noise ratio

(TSNR) image of study area is shown in Figure 5. As illustrated in Figure 5, the bridge and some buildings showed strong reflectivity with TSNR (more than 40 dB). The TSNR of the slip surface was relatively low, but it was basically greater than 20 dB. The stone revetment in the middle of the landslide and the stones stacked at the bottom of the landslide reached a TSNR of 30 dB. Figure 5b shows that the temporal coherence of the whole landslide area was between 0.8 and 1.0, which indicates that the quality of the acquired ground SAR image data is good [13,37].

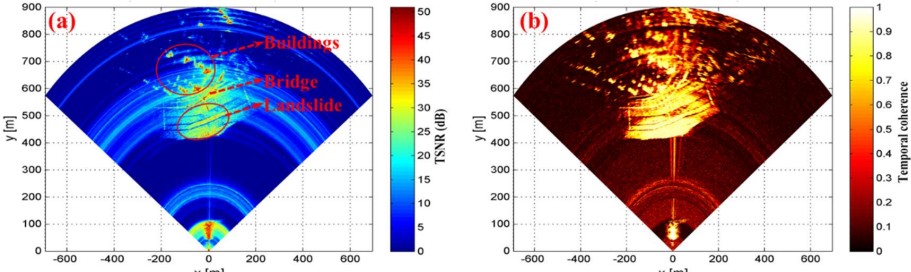

**Figure 5.** (**a**) The thermal signal noise ratio of study area. (**b**) The temporal coherence.

Such a high TSNR of study area enabled us to ensure a good estimation of the displacement sequence from 24 September to 6 October 2017, as shown from Figure 6.

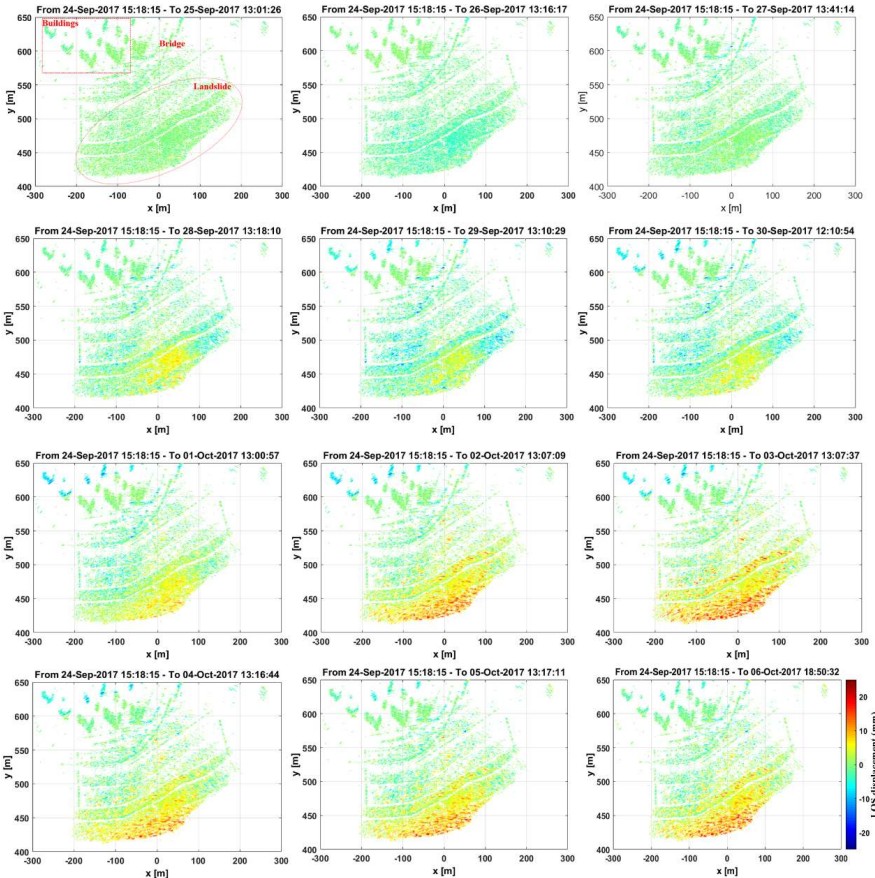

**Figure 6.** Interferogram sequence (Images acquired daily from 25 September to 6 October 2017 were processed with respect to the image acquired on 24 September 2017) showing the deformations in the Liusha Peninsula landslide. The white color corresponds to "no-data", where the data was eliminated in this analysis because of its low reflectivity. Note that negative displacement indicates approaching to the radar, whereas the positive displacement means away from the radar.

Figure 6 shows that the landslide had a nonlinear deformation during the 13-day monitoring period. As local landslide activity would be expected to cause positive changes in the LOS displacement, this was exactly the case of the lower part of the Liusha Peninsula landslide, where the positive displacement has increased. The maximum positive deformation value was 20.68 mm as of 6 October 2017. Unlike the lower part, the upper part of the landslide body was relatively stable. Furthermore, there was no evident deformation of the bridge during the whole period, which was basically in a stable state. Moreover, no obvious deformations of the buildings near the landslide were observed.

### 3.2. The Baishazhou Bridge

#### 3.2.1. Description of the Bridge and Experimental Setup

The Baishazhou Yangtze River Bridge is a double-tower and double-cable plane welded steel box girder cable-stayed bridge. The main beam adopts the mixed beam scheme: The steel box girder is used for the middle span and some side spans, and the concrete box girder is used for the two side spans. The beam is 3.0-m high and 30.2-m wide. It is basically a longitudinal floating system. A group of "elastic cables" are arranged from the lower crossbeam of each main tower to the left and right to anchor on the main beam 63-m away from the tower, so as to partially limit the longitudinal displacement of the main beam. Each cable surface is composed of 24 cables on the left and right, of which the cable spacing of concrete main beam is 24 m. The main tower is of diamond type with a height of 174.75 m, of which the part above the bridge deck is 145.0-m high, and the upper and lower beams are set. The following part of the bridge is mainly composed of six columns standing in the water, and the water depth is between 6–22 m. The geological column chart shows that the overburden of the riverbed is sand gravel layer, with a depth of 30–40 m.

Construction began on the bridge in in March 1997 and the bridge was opened to traffic in September 2000. The total length of the bridge is 3585 m, of which the length of main bridge is 2458 m and the main span is 1078 m (50 m+180 m+618 m+180 m+50 m). It has six double-directional driveways with a total width of 26.5 m and a capacity of 50,000 vehicles a day. Since its open to traffic in 2000, the bridge has undergone 24 repairs within 10 years. Therefore, dynamic monitoring campaigns of this bridge have been carried out since 2005 [45].

In this paper, high-frequency (10 Hz) GPS observations, lasting one hour, were conducted on 27 September 2016. In this campaign, two GPS receivers were set up upstream and downstream of the middle span (S012, S035), two GPS receivers were installed on the top of the double-tower (S035,S029), and the reference station was delineated, with the red triangle approximately 2-km northeast of the middle span (see Figure 7a,b). In addition, a GB-RAR monitoring of this bridge, lasting five hours, was also implemented using the IBIS-S system on 2 October 2017. The instrument was installed under the downstream deck with a sampling rate of 17 Hz. Moreover, two corner reflectors were fixed on the downstream bridge edge within the illuminated area of radar (Figure 7c), and one was mounted on a stable tripod 15 m in front of the radar, which was used to eliminate the atmosphere/noise effect (Figure 8).

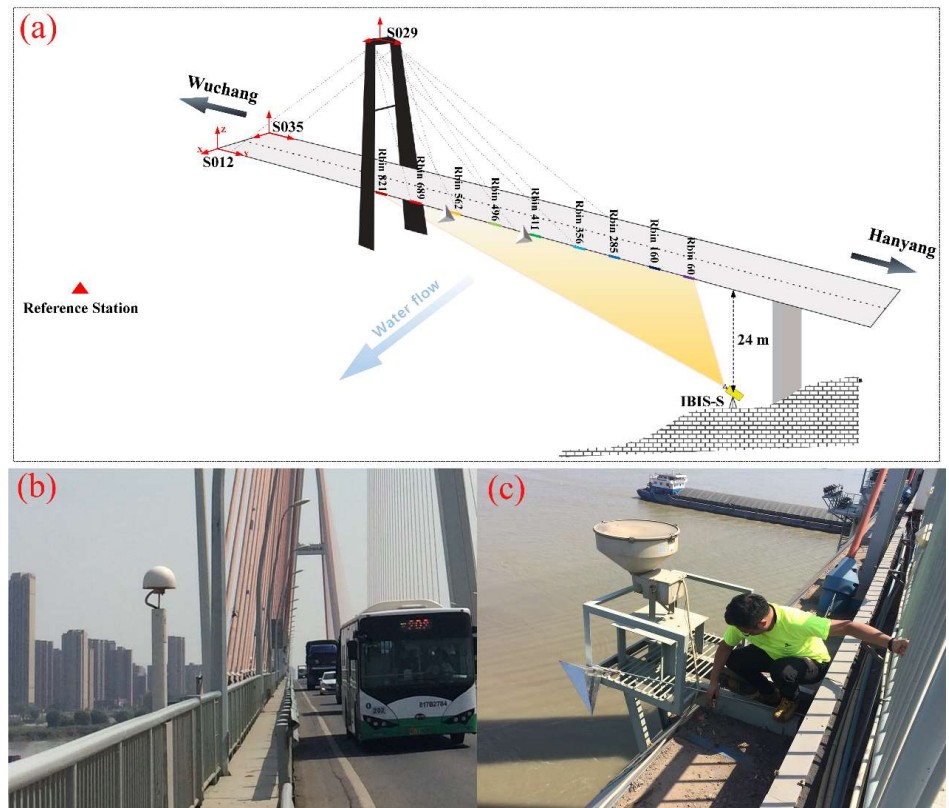

**Figure 7.** Sensors layout. (**a**) The diagrammatic sketch of GB-RAR measurement. Note that S023 is located on the East Bridge Tower, which is not shown in this sketch. (**b**) GPS observation. (**c**) Example of one of the corner reflector installations on the downstream side.

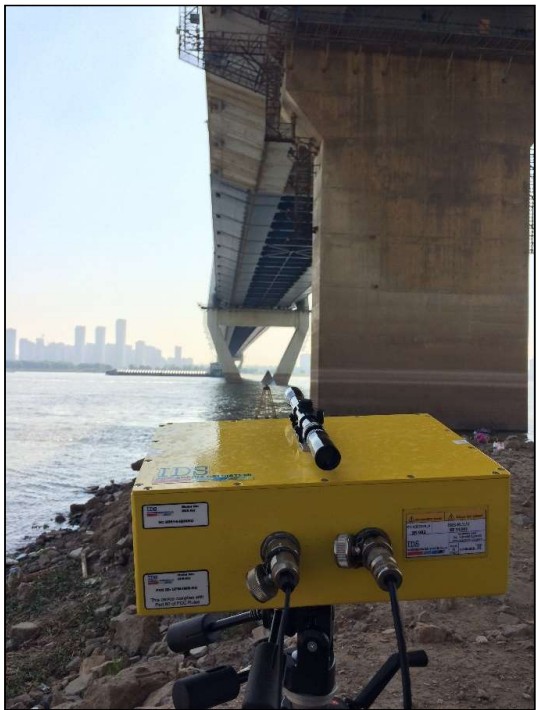

**Figure 8.** Field test of the ground-based real aperture radar (GB-RAR) to monitor the operational modes of the Wuhan Baishazhou Yangtze River Bridge.

### 3.2.2. Deflections of the Bridge

The operational test was carried out under the ambient excitation (usually due to micro-tremors, traffic loads, and wind) on the bridge. After acquiring a range profile of the deck bridge, we selected a set of radar bins with high TSNR from near to far along the deck and projected their LOS displacements to the vertical direction of the bridge according to their configure geometry. Due to the 17-Hz sampling rate and huge dataset, only 2000 s (34,000 records) displacement series of nine selected bins located at different distance along the deck were processed to extract the vertical vibration frequencies (Figure 7a). Meanwhile, the deflection sequences of the four GPS receivers in lateral (X), longitudinal (Y), and vertical (Z) directions under the Bridge Coordinate System (BCS) were transformed from the WGS84 coordinate system (N,E,U) by 2D similarity transformation [30]. Their corresponding vibration frequencies are shown in Figures 9 and 10, respectively.

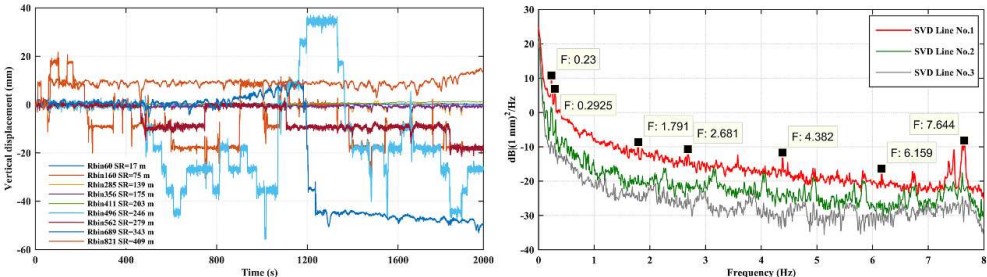

**Figure 9.** Vertical displacement time histories of different range bins along the bridge deck retrieved from radar measurements. The first three order SVD lines represent the vertical vibration frequency were extracted by the enhanced frequency domain decomposition (EFFD) method.

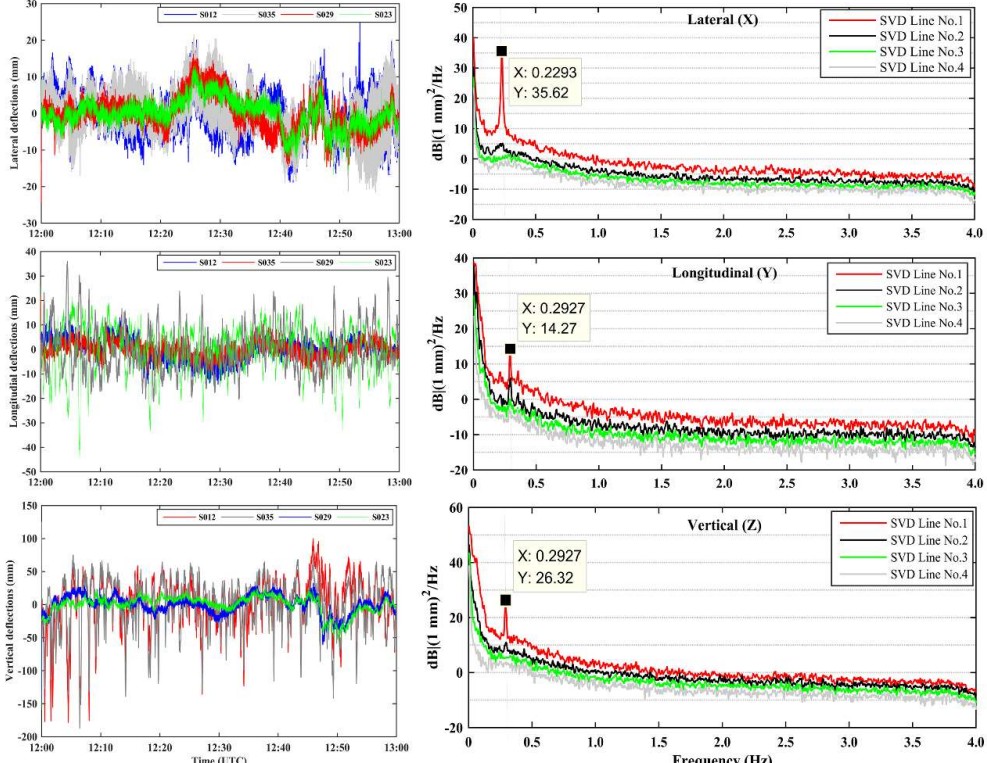

**Figure 10.** The three-dimensional deflection time histories under the BCS coordinates of S012, S035, S029, and S023 after removing mean values, and corresponding vibration frequencies in lateral, longitudinal, and vertical direction.

The analysis of the vertical deflections time histories retrieved from the GPS (Figure 10) and GB-RAR (Figure 9) indicates that the maximum deflections were more inclined to occur at the center of the bridge (e.g., Rbin689, S012). The downward deflection exhibited intermittent surge and rapid recovery characteristics. Meanwhile, the antisymmetric deformation characteristics of bridges (vertical direction anisotropic vibration at different positions) were also observed (e.g., Rbin496 and Rbin689 at 1200 s), which were expected as the excitation result triggered by the trucks rapidly approaching away from the middle deck. Although the GPS and GB-RAR measurements did not coincide with each other in time domain, it was impossible to make deflection cross-vibration of these two sets of measurements. However, for natural frequencies reflecting the inherent characteristics of the structure, dominant frequencies around 0.23 Hz and 0.29 Hz were derived from both vertical deflections of the GPS and GB-RAR (see Figures 9 and 10). Similar vibration frequencies were also retrieved by Zhang et al. [46] and Huang et al. [45]. The analysis involving the combination of multi-position and multi-direction (X, Y, and Z) deflection information of different sensors to implement the bridge modal parameters extraction and cross-validation are described in the Section 4.2.

## 4. Visualization, Cross-Validation, and Damage Detection

### 4.1. Web Browse of Landslide Deformation Based on VR Panoramic Technique

The results of deformation time series in the study area are usually expressed in radar two-dimensional coordinates or roughly superimposed on local DEM, which is not intuitive, especially when the displacement needs to be accurately projected onto specific objects in real scenes. Here, the three-dimensional development engine Unity was used to make the landslide deformation panorama and Web release according to the procedure in Section 2.2. Figure 11 shows the deformation panoramas of four important time events.

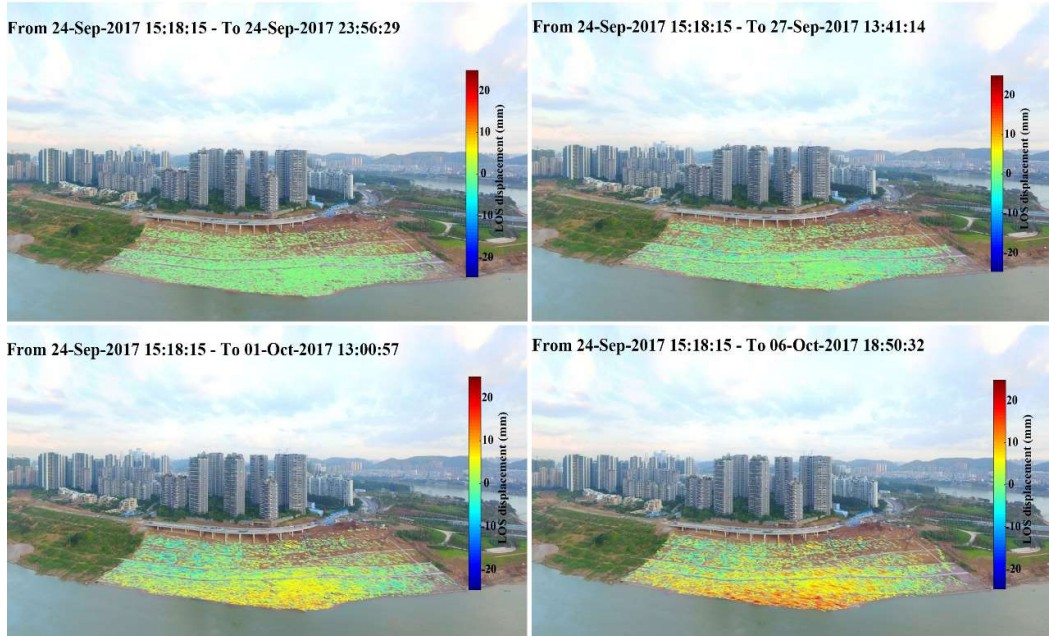

**Figure 11.** The LOS deformation panoramas sequence on 24 September, 27 September, 1 October, and 6 October 2017. The background is the 3-D terrain model of the study area produced from aerial photographs taken by the Dajiang Elves UAV.

As seen from Figure 9, the whole slope remained relatively stable before 27 September 2017. However, affected by several successive heavy rainfalls after 28 September 2017, the Liusha Peninsula slope showed obvious sliding. The deformation of the upper part was small, but the displacement of

the bottom of the slope reached 20 mm. Note that the bridge and the peninsula community located in the center and at the top of the slope, respectively, remained relatively stable during the whole observation period.

Furthermore, T2 and T3, with good quality of GPS observation data from 10:00 to 16:50, and 71 corresponding GB-SAR images were selected to compare their vertical displacement time series from the view of an individual point (Figure 12). The two stations were 50 m apart. Therefore, they showed the same trend of deformation in the six-hour observation and their cumulative displacements were less than 4 mm. However, due to the limitation of centimeter-level processing accuracy of high-frequency GPS, the displacements of the two stations were mostly constrained between 15 mm and –15 mm, which made it difficult to quantitatively compare the two types of displacements one by one, especially for the slowly deformed area around the Liusha Peninsula landslide.

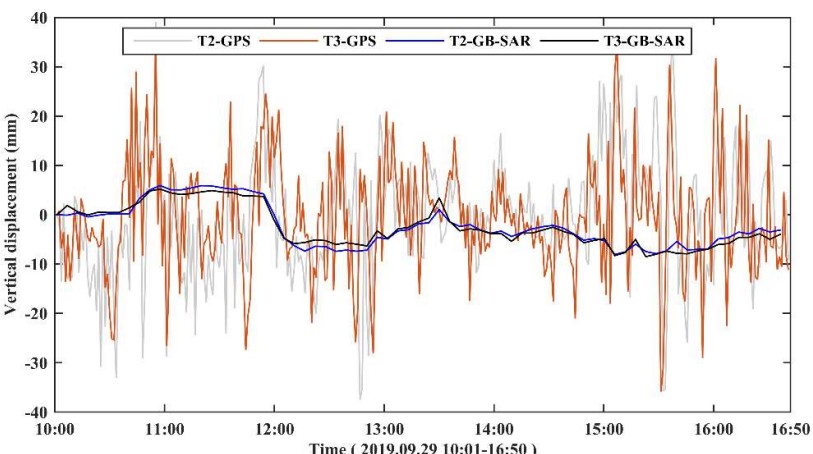

**Figure 12.** The vertical displacement time series of the ground-based synthetic aperture radar (GB-SAR) and GPS measurement.

## 4.2. Modal Parameters Extraction and Cross-Validation

Accurate identification of the first several resonance frequencies of bridge under natural environment excitation is considerably important to support and update numerical models or preventive diagnostic analyses [24]. With a relatively small structure rigidity of the long-span cable-stayed bridge, its natural frequencies of first several orders are low, generally at 0.1–5 Hz [45,47,48]. Therefore, the 10 Hz-GPS and 17 Hz-IBIS-S data sampling rates ensure the full extraction of natural frequencies. In this paper, the acquisition records of 2000 s and 3600 s corresponding to IBIS-S radar and the GPS receivers, respectively, were processed to extract the bridge's modal parameters using the EFDD method embed in ARTeMIS® (see Figures 9 and 10). Meanwhile, for the natural frequency, we used a hybrid optimization method (SQP-GA) that combined the genetic algorithm with sequential quadratic programming [24] to probe the dynamic vibration characteristics of the bridge. Next, we compared its vibration frequencies with results from the EFFD method (see Table 1). As seen from Figure 9, Figure 10, and Table 1, the amplitudes of different measuring points in the same direction were different, but the natural frequencies were basically the same. Compared with the more robust EFFD method, the frequencies derived from SQP-GA method showed little discrepancy with the EFFD-derived results, which indirectly proved the effectiveness of intelligent global search algorithm in extracting the modal parameters of large structures. Moreover, compared with GPS in-situ measurement, the radar measurement method can not only detect multiple point displacements at the same time, but also detect higher order vibration frequencies of bridges due to its high sampling rates, such as 1.791 Hz, 2.681 Hz, 4.382 Hz, and 7.644 Hz (Figure 9).

**Table 1.** Full-test results of the multi-frequency extraction of the two sensors based on the EFFD and SQP-GA methods.

| Direction | Sensor | Method | Main Frequency (Hz) | | | |
|---|---|---|---|---|---|---|
| vertical | GPS | EFFD | – | 0.2927 | 0.3801 | – |
| | | SQP-GA | – | 0.2927 | 0.3724 | – |
| | GB-RAR | EFFD | 0.23 | 0.2925 | 4.382 | 7.644 |
| | | SQP-GA | 0.2293 | 0.2921 | 4.212 | 7.402 |
| Lateral | GPS | EFFD | 0.2293 | – | – | – |
| | | SQP-GA | 0.2291 | – | 0.35 | – |
| Longitudinal | GPS | EFFD | – | 0.2927 | 0.3561 | – |
| | | SQP-GA | – | 0.2927 | 0.3721 | – |

In order to distinguish among bending and torsion modes during the frequency analysis of the recorded data, there must be a phase reference between the data recorded on the upstream and downstream side of the bridge. To this end, the radar must be installed on a platform that can simultaneously illuminate the upstream and downstream decks of the bridge. However, it is not always the case that this solution can be applied to all bridges, since the visibility of the objects located on two sides is limited by the angular resolution of IBIS-S and the thickness and width of the bridge deck. This is the exact case of the Wuhan Baishazhou Yangtze River Bridge. Therefore, we usde the one-hour records with 12 channels of four GPS receivers (S012, S035, S029, and S023) to extract its dynamic vibration parameters.

Figure 13 shows the identified vibration modes. The first lateral flexural mode shape was symmetric, with a frequency of 0.217 Hz (Figure 13a), while the second and third recognition modes (0.287 Hz and 0.38 Hz) involved the vertical bending of bridge decks constrained by the east and west bridge towers. In this case, the third mode shape presented the same vibration characteristics as the second mode but with low amplitude. Therefore, only the second mode shape (0.287 Hz) was sketched here (Figure 13b). Figure 13c,d show the MAC correlation of the first three modes, in which the self-power-spectrum correlations of each modes were strong but the cross-spectral correlation were weak. However, affected by sampling rate (10 Hz) and kinematic processing accuracy (cm) of GPS measurements, the higher modes involved torsional mode shape were not identified. However, they were detected by IBIS-S with high sampling rate (17 Hz, and even more) and submillimeter measurement accuracy (e.g., 1.791 Hz, 2.681 Hz, 4.382 Hz, and 7.644 Hz). In order to take full use of these advantages of the GB-RAR technique—and therefore, in order to make full use of these advantages of GB-RAR measurement to capture the vibration modes of bridges globally—we need to consider the following problems in the next work: (1) By installing multiple corner reflectors on the upstream and downstream of bridges, the deflection at different positions on both sides of the bridge can be accurately recorded; (2) By setting up two IBIS-S systems underneath different positions of the bridge, the true three-dimensional vibration characteristics of the bridge are extracted. Subsequently, it is possible to extract multi-order mode shapes of bridge accurately and comprehensively based on the high-precision three-dimensional deflection time series of a large amount of measuring bins located at different positions of upstream and downstream bridge deck.

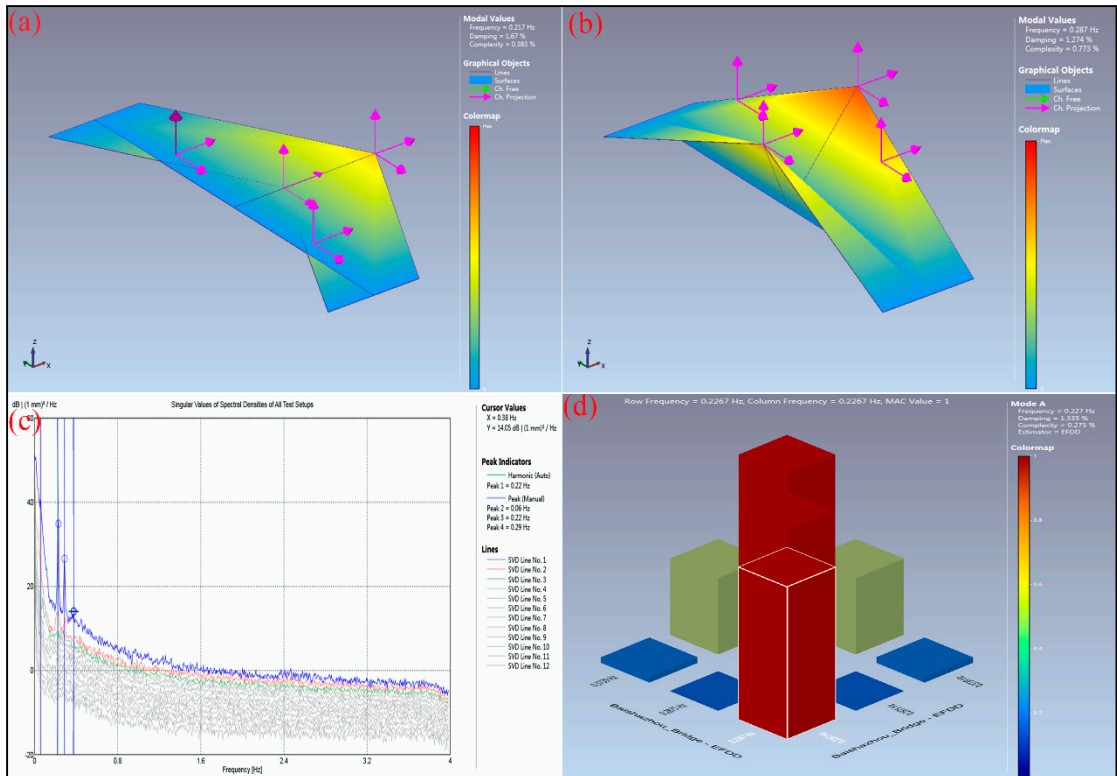

**Figure 13.** Vibration mode shape and corresponding modal assurance criterion diagram (MAC) correlation matrix of the bridge using the EFFD technique from the GPS measurement. (**a**) F = 0.217 Hz, Damping = 1.67%. (**b**) F = 0.287 Hz, Damping = 1.27%. (**c**) First three peaks, i.e., 0.22 Hz, 0.29 Hz, and 0.38 Hz. (**d**) MAC correlation of the three peaks.

## 5. Conclusions

Two case studies based on ground-based radar interferometry were described here, i.e., the Liusha Peninsula landslide and Baishazhou Yangtze River Bridge, which confirmed the capability of ground-based radar interferometer to detect the real-time rapid deformation field of landslide sectors and completely operational modal parameters of bridges. For the Liusha Peninsula landslide, we obtained the daily evolution time series of displacements, with a maximum cumulative displacement of 20 mm in the 13-day observation period. The GPS observation provided a rough comparison between the displacement derived by the noncontact and in-situ measurement, showing that the vertical deformation trends of some specific points (T2, T3) were generally stable. Furthermore, the displacement evolutions of the landslide were overlying on a spherical panorama images, which enabled nonprofessionals to locate local deformation intuitively. For the Baishazhou Yangtze River Bridge, the GB-RAR technique can not only extract lower natural frequencies, just as the results derived from GPS (e.g., 0.23 Hz and 0.29 Hz), but can also detect higher natural frequencies (e.g., 1.79 Hz, 2.68 Hz, 4.38 Hz, and 7.64 Hz), that may involve torsional modes not detected by the GPS technique. The consistencies of frequency results derived from SVD-based EFFD method and global optimization-based SQP-GA algorithm indicate the potentiality of using noncontact GB-RAR technology to obtain accurate deflections and modal parameters of bridge under the operation state.

However, the GB-RAR technique only provided a 1-D LOS displacement and failed to capture the lateral vibration of the bridge. To overcome this problem in future work, multiple corner reflectors should be properly installed on the upstream and downstream decks of bridges. At least two GB-RAR systems should be set up properly at a place where they can simultaneously illuminate the corner reflectors mounted on the upstream and downstream decks of the bridge. Subsequently, it is possible

to obtain precise 3-D deformation of different points located on the upstream and downstream deck and implement the health monitoring of the bridge comprehensively.

**Author Contributions:** This paper is a collaborative work by all the authors. J.H. conducted the experiments, programmed the algorithm, and wrote the manuscript. J.G. contributed the idea and organized the projects, L.Z., Y.X. and K.F. participated the experiments and processed the data; S.Z. assisted with certain experiments, and all authors proof-read the paper.

**Funding:** This research was sponsored by the National Natural Science Foundation of China (No. 41474004, 41704002), the Open Fund of Guangxi Key Laboratory of Spatial Information and Geomatics (Grant No. 17-259-16-03) and Fund of Nanning Natural Resources Information Center.

**Acknowledgments:** This work has also been greatly helped by Xinpeng Wang and Ruijie Xi from Wuhan University. We would like to thank the editor and the anonymous reviewers for their valuable comments, which greatly improved the quality of this manuscript.

**Conflicts of Interest:** The authors declare no conflict of interest.

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
