# Peer review of "Differential Ground-Based Radar Interferometry for Slope and Civil Structures Monitoring: Two Case Studies of Landslide and Bridge"

_remotesensing, doi:10.3390/rs11242887_

Round 1

Reviewer 1 Report

In this paper, GB-RAR and GB-SAR that provide great deformation resolution with mm are used to evaluate the deformation of a landslide and a bridge, respectively. I believe that the results and approaches of this paper should be interested for researchers of related field and reader of Remote Sensing. I strongly recommend this paper can be published in this journal after some modifications. Some questions and comments are listed below.

GB-RAR and GB-SAR provide very high accuracy of deformation, so that authors need illustrate how to separate background noise induced by different environmental conditions and real deformation information. Furthermore, authors need also demonstrate variations of deformation information obtained in different environmental conditions. More descriptions about how to identify the studied landslide by using data in Fig. 5 are necessary. In my opinions, bulging at the toe of a slowing moving landslide is a common feature. In this studied case, the toe area is approaching toward the observer that indicates the LOS displacement is negative. However, the paper mentioned that the monitoring data show a positive LOS displacement. Unless some erosion occurred at the toe area so that, a positive LOS displacement can be observed. So, please check the data or the interpretation. In addition, a conceptual model of deformation of landslide will help readers to understand the monitoring data. Rainfall information is important to verify the results of GB-SAR and GPS. If it is possible, the author should provide the rainfall data at or nearby the studied site. Furthermore, authors can using GPS and rainfall data as constrain to discuss the MT In-SAR data. Authors need explain why data in Fig 9 shows significant variations in vertical displacement at some places.

Author Response

Thank you very much for your recommendation of this paper published in RS journal.

Reviewer 2 Report

The article describes the application of two synthetic aperture radar methodologies (GB-SAR) and real aperture ground radar (GB-RAR) for monitoring a landslide (Liusha peninsula landslide Yong River in Nanning) and a bridge (Baishazhou Yangtze River Bridge). The case studies are well presented only with regard to the application of the technique which does not represent a novelty in the field of research and data acquisition. The techniques applied are different and as valid as the data even though they deal with different problems that have not been fully addressed. The work shows a good acquisition of data with different techniques on two different issues but not comparable as cases of studies. Good comparison with GPS data. In particular, the work is completely lacking in the interpretation of the data. For the landslide there is a complete lack of analysis of the geological and geomorphological context and the description of the type of landslide that affects the area. This would make it possible to highlight whether the movements and therefore the observed deformation is compatible with the type of movement expected for a given classification of landslide. As an example it would be interesting to verify if the deformation shown in Figure 6 and in the work follows the oscillations of the water level in the body of the landslide and simultaneously of the river itself. The same goes for the bridge. There is a complete lack of structural characterization of the work (type of construction, type of foundations, type of land on which the work insists) that helps to assume what the monitoring data represents.Therefore, although the acquisition techniques are valid, there is no interpretation, even if only probable, of the fact that it does not fall on the landslide problem and the bridge problem. It is also advisable to deal separately with the topics in different articles that show the application of different techniques on case studies of landslides and on case studies of bridges with a solid interpretation of the data and not only as a simple collection of displacement data. The comparison with GPS data and the bibliographic treatment is good. Therefore, the article cannot be published in the present form. The authors are invited to deepen the two problems with a solid analysis and interpretation of the displacement data that is still valid and of quality.

Author Response

Thank you very much for your strict suggestions for improvement.

Reviewer 3 Report

Journal: Remote Sensing (ISSN 2072-4292)

Manuscript ID: remotesensing-631337

Title: Differential ground-based radar interferometry for slope and civil structures monitoring: two case studies of Landslide and Bridge

Authors: Jiyuan Hu, Jiming Guo, Yi Xu, Lv Zhou, Shuai Zhang, Kunfei Fan

Comments and Suggestions for Authors

The Authors present a paper focused on the use of Ground-based radar interferometry for monitoring the movement of landslide and man-made structures as a Bridges. Two case study are presented related to the daily monitoring of Liusha peninsula landslide and the Baishazhou Yangtze River Bridge in China, considering the virtual reality-based panoramic technology (VRP) in order to intuitively illustrate the displacement evolutions within a web-based panoramic image browsing. Full-scale test and time-frequency analysis using two different monitoring methods (ground-based Real Aperture Radar GB-RAR) and GPS were performed, highlighting the potentiality of the use the ground-based radar in the bridges health monitoring. Title and content of the paper are according to the scope of the journal. The whole structure of the work is a well-written with well-defined goals that are achieved and explained in the results. In order to improve the manuscript before of its publication, only few suggestion focused on the literature review should be considered by the authors in the first part of the document while revising the paper. In particular:

- Page. 1 (line 31 - Introduction section): The authors speak about "precision" of SAR deformation measurement. The precision is generally referred to the acquisition system through the comparison with another or similar acquisition techniques. If the authors refers to the value of the measure (i.e. the defamation, which in the case of SAR acquisition is expressed with an annual average velocity or cumulative displacement in the observation period, would be more correct to speak about "accuracy". I suggest to replace the word "precision" with "accuracy" and add some reference present in literature such as (Casu et al. 2006; Nicodemo et al., 2017,.....).

- Page. 1 (line 31 - Introduction section): Only a few references have been mentioned by the authors about the use of SAR data in different application fields ([1] subsidence, [2] earthquake and [3] landslide). In the last few years, data acquired from SAR systems and processed via differential interferometry techniques were used in several application related to both natural and anthropogenic hazards. So, for a more complete scientific information, other references should be added by the authors as: Cigna et al., 2012; Peduto et al., 2017a for subsidence; Prati et al., 2010; Reale et al., 2011 for earthquake; Wasowski and Bovenga, 2014; Peduto et al., 2017b, for landslide; Giannico et al., 2012 for underground construction works; Perissin et al., 2009; Peduto et al., 2017b for structures/infrastructures monitoring.

Casu F., Manzo M., Lanari R. (2006). A quantitative assessment of the SBAS algorithm performance for surface deformation retrieval from DInSAR data. Remote Sensing of Environment, 102(3-4):195-210.

Nicodemo G., Peduto D., Ferlisi S., Maccabiani J.(2017). Investigating building settlements via very high resolution SAR sensors. In: Bakker J. et al. (eds.), Life-Cycle of Engineering Systems: Emphasis on Sustainable Civil Infrastructure. Proc. of the Fifth International Symposium on Life-Cycle Civil Engineering (IALCCE 2016), Taylor & Francis Group, London, pp. 2256–2263, (ISBN 978-1-138-02847-0).

Cigna F., Osmanoglu B., Dixon T.H. De Mets C., Wdowinski S. (2012). Monitoring land subsidence and its induced geological hazard with synthetic aperture radar interferometry: a case study in Morelia, Mexico. Remote Sens. Environ., 117:46–161.

Peduto D., Nicodemo G., Ferlisi S., Maccabiani J. (2017a). Multi-scale analysis of settlement-induced building damage using damage surveys and DInSAR data: a case study in The Netherlands. Engineering Geology, 218:117–133.

Prati C., Ferretti A., Perissin D. (2010). Recent advances on surface ground deformation measurement by means of repeated space-borne SAR observations. J. Geodyn. 49:161–170.

Reale D., Fornaro G., Pauciullo A., Zhu X., Bamler R. (2011). Tomographic imaging and monitoring of buildings with very high resolution SAR data. IEEE Geosci. Remote Sens. Lett. 8, 661–665.

Wasowski J. and Bovenga F. (2014). Investigating landslides and unstable slopes with satellite multi temporal interferometry: current issues and future perspectives. Eng. Geol. 174:103–138.

Giannico C., Ferretti A., Alberti S., Jurina L., Ricci M., Sciotti A. (2012). Application of Satellite Radar Interferometry for Structural Damage Assessment and Monitoring. In: Strauss A. et al. (eds.), Life-Cycle and Sustainability of Civil Infrastructure Systems. Proc. of the 3rd International Symposium on Life-Cycle Civil Engineering (IALCCE 2012), CRC Press, Boca Raton, FL, pp. 2094–2101.

Perissin D., Prati C., Rocca F. (2009). PSInSAR analysis over the Three Gorges Dam and urban areas in China. Urban Remote Sensing Joint Event 2009, 1–5.

- Page. 221 (line 31 - 3.1.2 Displacement results section): Please improve the readability of the figure by increasing, for instance, the size of the legend fonts and the titles along the x-y axis as well as the acquisition time at the top of the graphs.

Author Response

Thank you very much for your high approvement and detailed suggestions.

Reviewer 4 Report

The work is interesting but it is not well described especially in what is the original part of the paper.

In fact, the authors dedicate several space to the description of radar monitoring, whose the literature is rich, while the description of the innovative topics is too brief.

To elaborate on this aspect, I consider a major revision appropriate.

The English is not completely perfect or good because of some grammar mistake as the “s” at the third person and correlation between the subject and the verb.

In some parts the periods are too long.

When expressing a particular day, it is better to use the ordinal numbers. Example 6th October 2017 instead of 6 October 2017.

Line 56

I advise the authors also to add the monitoring of the earth dams, citing the following paper:

Di Pasquale, A.; Nico, G.; Pitullo, A.; Prezioso, G. Monitoring Strategies of Earth Dams by Ground-Based Radar Interferometry: How to Extract Useful Information for Seismic Risk Assessment. Sensors 2018, 18, 244.

The authors repeat throughout the text (see Figure x). It would be advisable to delete “see.”

Figure 8. In the text only Figures 8-a is indicated. Indeed, Figures 8-b and Figure 8-c are not indicated.

Line 50

which by this way can realize = which can realize

Line 60

e.g., 1) = e.g.: 1)

Line 64

professionals. 2) = professionals; 2)

Line 71

Subsequently the = Subsequently, the

Line 76

With these damage-sensitive features = Through these damage-sensitive features

Line 92

This paper = In this paper

Line 98

Thereby it can = Thereby, it can

Line 99

thus lead to = thus leading to

Line 113

(3),and =  (3), and (put space after the comma)

Line 122

the phase discrepancy during a time interval of the same range bin are obtained = the phase discrepancy during a time interval of the same range bin is obtained (or the phase discrepancies during a time interval of the same range bin are obtained)

Line 123

in Equation (5), which = in Equation (5) which

Line 127

eliminate = eliminated

Line 130

But the tripod = The tripod

Line 163

is the i-th matrix of FRF; = The subject is missing

Line 186

2014 due to = 2014, due to

Line 228

the case in the lower part = the case of the lower part

Line 228

The bridge started construction = The bridge started its construction

Line 249

October 2, 2017 = October 2nd, 2017

Line 300

four important time event = four important time events

Line 308

Note that= It can be noted

Line 323

is of considerably important = is considerable important

Line 368

GBRAR technique, Therefore, = GBRAR technique, and therefore,

Line 383

The GPS observation provide = The GPS observation provides (or The GPS observations provide)

Author Response

Thank you so much for reading this manuscript so carefully.

Round 2

Reviewer 2 Report

The authors' responses are considered satisfactory. Authors are advised to report and better describe in the article what they themselves wrote in the points:

In abstracts and conlusions

1: However, in this manuscript for landslide, we emphasized the 3-D display by combining the GB-SAR results with VRP, while for the bridge, we care more about the no-contact resonance vibration frequency and MAC, as well as the corresponding detection methods.

In the discussions highlight the response in point 2 for the Yongjiang River landslide. Starting from the bibliography mentioned above, describe the simplified geological and geomorphological characteristics of the slope, as well as the characteristics of the bridge. Always justify in the text the lack of possibility of comparison between water levels of the river and movements of the slope.

The answer to point 4 is considered acceptable.

Therefore, in view of the modifications and the answers, the authors are asked to proceed with a further descriptive enrichment on the landslide, taking inspiration from the contents of the bibliographic quotations that they themselves suggest. This further modification is considered necessary for the publication of the work.

Author Response

Thank you very much for your approval. I tried my best to correct the manuscript as what you suggested. And for some points, I really can't make too deep explanation currently because of lacking geological data. I hope you can see my attempt and effort in the expression of Surveying and mapping results. Thank you again!

Reviewer 4 Report

The authors have responded to most of my observations but I also wanted an extension to paragraph 2.2. “VR-based Panoramic Technology of web” where the procedure is indicated only for points. Perhaps, I have not been clear in my previous review.

Even in paragraph 4.1, which is given the title “Web browse of landslide deformation based on VR Panoramic technique”, there are only a few lines that reflect the title of the paragraph while everything else is dedicated to comments on GBSAR and GPS data. The same applies to the conclusions.

I reiterate that the work is interesting and, therefore, suitable for publication after minor revision, since it would be more complete if the authors would better describe the aforementioned paragraphs.

Other comments:

Reference [26]

Authors have included, in the new version, the reference [26] suggested by me, but I don't understand why they wrote the reference [26] in the wrong way (reversing the authors' last names with their names), although I have suggested the correct way.

Therefore, it is necessary to correct the reference [26] which must be written as follows:

Di Pasquale, A.; Nico, G.; Pitullo, A.; Prezioso, G. Monitoring Strategies of Earth Dams by Ground-Based Radar Interferometry: How to Extract Useful Information for Seismic Risk Assessment. Sensors 2018, 18, 244.

https://doi.org/10.3390/s18010244

The authors did not accept this comment of mine and, consequently, did not correct the dates:

“When expressing a particular day, it is better to use the ordinal numbers. Example 6th October 2017 instead of 6 October 2017”.

I continue to be convinced that the dates should be written this way, 6th October 2017, and not in this 6 October 2017 because the first method is used in UK English, while the second in US English: to verify.

In some cases, the authors claim, in the note, to have made the correction but in the text (new version of the paper) the correction was not made (I guess simply for distraction), for example:

Line 170

is the i-th matrix of FRF; = The subject is missing

Authors' reply in the cover letter: Ri is the i-th matrix of FRF;

In the text of the paper (new version): is the i-th matrix of FRF;

Line 394

techniqueand therefore, = technique, and therefore,

Some small things to correct:

Line 114

(3)and = (3) and (put space after the comma)

Line 127

d,in line = d, in line (put space after the comma)

Author Response

Thank you very much for your detailed correction and continuous persistence, as well as your extension suggestions.
